# Assessing Executive Cognitive Functions in Sheep: A Scoping Review of Recent Literature on Cognitive Tasks

**DOI:** 10.3390/ani15243647

**Published:** 2025-12-18

**Authors:** Davide Galanti, Emanuela Dalla Costa, Sara Barbieri, Michela Minero

**Affiliations:** Department of Veterinary Medicine and Animal Science (DIVAS), University of Milan, 26900 Lodi, Italy; emanuela.dallacosta@unimi.it (E.D.C.); sara.barbieri@unimi.it (S.B.); michela.minero@unimi.it (M.M.)

**Keywords:** sheep, cognition, cognitive tasks, cognitive functions, brain processes, animal welfare neuroscience

## Abstract

Understanding how sheep memorize and learn can help both scientific research and farm management. This review looked at how researchers have tested cognitive skills in sheep over the past decade, including memory, learning, problem-solving, and decision-making. After searching scientific databases, 20 studies were thoroughly reviewed. These studies used different tools and setups to test how sheep respond in certain situations, such as finding rewards or adapting to new environment. The review showed that certain executive cognitive functions are closely linked to specific task designs. The tests were designed to be as stress-free as possible for the animals. Overall, learning more about how sheep think and solve problems can improve animal care, support better translational research, and offer insights that may help transferring results to human medicine.

## 1. Introduction

The study of cognitive abilities in animals has evolved considerably, encompassing a wide range of species, including livestock such as sheep (*Ovis aries*). Cognitive functions in animals refer to mental processes like memory, perception, learning, and decision-making [1], which can significantly influence their behavior and welfare [2]. In sheep, understanding cognitive abilities is essential not only from a biological and ethological point of view, but also for practical implications in animal husbandry and welfare. Historically, sheep have been primarily studied for their productive and reproductive traits to optimize production. However, recent research has highlighted the importance of cognitive traits in influencing sheep behavior and welfare [3]. Sheep are versatile animals with numerous advantages for research and agriculture. Their docile nature, ease of breeding, and adaptability to various climates make them ideal for study and husbandry [4]. Sheep provide a triple-purpose use through meat, milk, and wool production, meeting diverse human needs. Numerous studies have enhanced our understanding of sheep, particularly in milk production, allowing for improved selection criteria and conservation of diversity, also related to cognition and temperament [5,6]. Furthermore, research on sheep docility has revealed its impact on reproductive success [7]. Studies have also shown that sheep possess complex cognitive abilities, individual personalities, and social behaviors, challenging the stereotype of them being unintelligent [8]. Lastly, health reflects the physiological and neurological capacity to respond to disease and maintain balance, making it an essential component of welfare rather than a distinct condition. All these aspects are strongly interconnected [2].

Cognitive tests can help identify individual differences in learning, memory, and problem-solving abilities, which can be crucial for improving management practices, breeding programs, and welfare standards in sheep [9]. Due to its convoluted encephalon, extended lifespan, and natural cognitive decline, sheep offers a useful alternative to murine models [10]. Because executive functions have been extensively characterized in mice for decades, murine studies provide a well-established benchmark for evaluating and validating similar cognitive paradigms. Executive cognitive function in mice can be assessed using a variety of validated behavioral tests that probe attention, flexibility, inhibitory control, and memory. The puzzle box test efficiently evaluates problem-solving and executive functions in different mouse models of schizophrenia [11]. Touchscreen-based tasks, including extinction, reversal learning, and the five-choice serial reaction time task, measure distinct aspects of executive function such as inhibitory control, cognitive flexibility, and sustained attention [12]. The German Mouse Clinic employs a comprehensive test battery including spontaneous alternation in the Y-maze, social discrimination, object recognition, and olfactory discrimination learning to provide an integrative assessment of cognitive performance [13]. The Y-maze derives its name from its three arms radiating from a central point at 120° angles, resembling the letter “Y” [14]. It is commonly used to evaluate spatial working memory by exploiting a mouse’s natural tendency to explore novel environments. The T-maze, shaped like the letter “T,” consists of a straight start arm intersecting perpendicularly with two choice arms. This design is frequently employed in decision-making and reward-based learning studies, allowing researchers to assess spatial learning, reversal learning, or preference [15]. Moreover, the testing arena refers to the controlled experimental space in which trials are conducted, typically enclosed to prevent escape and equipped with visual or tactile cues to guide navigation [16]. The problem-solving setup encompasses the arrangement of barriers, cues, or reward locations designed to require the subject to employ memory, strategic decision-making, or cognitive flexibility to achieve a goal [17].

Most of these paradigms have also been described in several species, including sheep (Figure 1). The interest in studying cognitive functions in sheep comes from various motivations: firstly, enhancing animal welfare is a paramount concern in modern animal husbandry. Cognitively enriched environments and knowledge related to the mental capabilities of sheep can lead to better welfare practices by reducing stress and promoting natural behaviors [18]. Secondly, cognitive testing in sheep can improve management practices. For example, understanding how sheep perceive and interact with their environment can inform better handling and housing practices, avoiding fear and other negative stimuli [19]. Moreover, there is growing interest in using sheep as a preclinical models to investigate human neurological and cognitive conditions, offering valuable insights for translational research [20,21]. This comparative aspect can provide insights into the evolution of cognition and brain function across species.

Given the rising interest and expanding body of research on sheep cognition, it is crucial to evaluate and synthesize the existing evidence to draw conclusions and provide reliable recommendations. Conducting a systematic review to address a practical problem represents the highest level of evidence in the hierarchy of evidence-based medicine. When properly executed, a systematic review offers a rigorous and replicable method for identifying, evaluating, and summarizing existing evidence. A scoping review is a distinct but equally valuable form of knowledge synthesis. It is typically conducted to map and summarize the available literature on a given topic, and also when a systematic review is not feasible; such as when the evidence base includes studies with diverse designs, methodologies, or objectives [24,25]. It can work as exploratory research, providing direction to future research by identifying knowledge gaps [26]. This approach not only enhances the comparability and reproducibility of results but also enforces the evidence base, ensuring that recommendations for standardization are grounded in comprehensive and methodologically sound research. The standardization of cognitive tests in sheep research is vital, as the observed variability in methodologies across studies can block comparability and reproducibility [27]. Standardization can facilitate the development of reliable, comparable, and valid tests that can be consistently applied across different research contexts [28], thereby enhancing the robustness of findings and their applicability to practical settings.

This scoping review aimed to identify, classify, and describe the cognitive tasks designed to assess different executive cognitive functions in sheep, such as memory and flexibility, with the goal of helping standardize their use in neuroscience research.

## 2. Materials and Methods

A literature search, based on the methodology outlined in the Preferred Reporting Items for Systematic Reviews and Meta-Analyses (PRISMA) guidelines [29], was conducted through three electronic databases: CAB Abstracts, PubMed, and Scopus. The search aimed to identify the available literature on cognitive tasks designed to assess various cognitive functions in sheep, encompassing all papers published between the 1 January 2010 and 4 August 2025. The vast majority of detailed test descriptions have been published in recent decades. Earlier studies on sheep cognition are acknowledged as foundational to the field; however, they fall outside the scope of the present review, which focuses on more recent research. The search strategy employed the following word string, performed within the title, abstract, and keyword fields:


*(Sheep OR Ovis OR Lamb) and (Cogniti * OR Memory OR Flexibility OR Choice) and (Task OR Test OR Maze OR Image OR Reward).*


All scientific papers obtained from the database search were compiled into an Excel spreadsheet (Microsoft Office) for organization and screening. The screening process was conducted in multiple sequential stages. At the first stage, studies were screened based on the following three criteria: (1) articles written in English; (2) availability of an abstract and clearly identified authors, to ensure sufficient information for initial evaluation; (3) original articles. Duplicates were removed. In the second phase of screening, studies were evaluated based on the presence of specific keywords. Abstracts were assessed for the terms “sheep,” “ovine,” or “lamb”, while author-provided keywords were checked for “sheep,” “cognition,” “model,” “ewes,” and “neuro”. These keywords were selected to focus the review on studies assessing cognitive functions in sheep. This step allowed for a rapid identification of potentially relevant articles while maintaining focus on the review objectives. Subsequently, a more in-depth review of each title was carried out to determine whether the manuscript was relevant and pertinent. All selected articles were screened at the abstract level, and, when necessary, the Materials and Methods section was examined to ensure that the cognitive tasks were specifically defined, in dimensions and functioning. As a final step, articles were excluded if they met any of the following criteria: (1) lack of a detailed description of a cognitive test or absence of explicit information on the test arena or apparatus design; (2) focus on “emotional cognition” as the primary cognitive domain investigated; (3) inclusion of miscellaneous topics unrelated to the objective of the review; (4) duplicate reports using the same animals and experimental setup. When these criteria were not clearly evaluable by the abstract, articles were subjected to full-text screening. The entire screening process was performed independently by the first author (D.G.) and the second author (E.D.C.) to ensure accuracy and consistency. In cases of disagreement, both authors jointly reviewed the studies and reached a consensus on inclusion or exclusion. Data from the selected papers were summarized in different tables to highlight the executive cognitive function studied. The table also included animals’ identifying data such as category and sex, the type of maze or arena, its design and functioning.

## 3. Results

The initial literature search retrieved 797 from CAB Abstracts, 1006 from PubMed, and 1070 records from Scopus, for a total of 2873 records. After the first screening step, a total of 1740 records were discarded. Following this, an additional 1720 papers were removed in the second step. Details regarding the number of excluded studies and the reasons for their exclusion are outlined in a modified PRISMA flow diagram (Figure 2).

Ultimately, 20 papers were selected for inclusion in this review and they were analyzed in relation to the executive cognitive functions investigated and types of tasks applied. The dimensions and configurations of the maze or arena used in the studies were also elaborated and discussed in greater detail. Table 1 provides a summary of the key features of the included studies, such as the research focus, the typology of test, and the animals examined (including their category and sex). In all studies, the cognitive function investigated was explicitly stated as the main objective of the authors’ work.

### 3.1. Executive Cognitive Functions

The cognitive functions of sheep pertain to abilities not related to emotionality, human–animal or mother-offspring bonds, which have already been analyzed by various authors [19,46,47,48]. In this case, the inclusion criteria focus was directed on functions more closely associated with spatial memory, sensory, and adaptability processes, with particular attention to sheep of both sexes and various age categories [49,50]. The executive cognitive functions examined in the included literature were grouped into three main categories: cognitive flexibility and memory; sensory discrimination and association; decision-making and problem-solving. Five out of the 20 studies addressed multiple cognitive domains [10,30,31,32,42]. During screening, nearly all of the included papers were found across all three databases.

Table 2, Table 3 and Table 4 provide detailed information about the type of test, the presence or absence of a pre-test period, the size and construction of the testing apparatus, as well as its functioning. These tables have been divided based on the executive function that was reported to be assessed by the authors of the papers included. Through the analysis, it emerged that certain studies, both recent and older, are frequently cited as methodological references by other authors [23,51]. We observed that the majority of mazes were based on previous papers; all problem-solving setups were based on previous work; regarding arenas, most were newly designed, except for the McBride setup [23], which has been used several times in subsequent studies.

#### 3.1.1. Cognitive Flexibility and Memory

Twelve studies out of 20 address cognitive flexibility and memory (Table 2). To investigate these functions two modified Y-maze systems [10,30], two T-mazes [38,42], and three spatial mazes [9,41,43] were used. Various arena designs were used, including semi-automated operating systems for a two-choice task [31], a two-choice discrimination task [39], a spatial A-not-B detour task [32], and a food acquisition task in the arena [40], each with its specific features. One problem-solving set-up with cylinders was also mentioned [34]. Regarding the pre-test period, three studies did not describe it [39,42,43], three included only a habituation phase [9,30,32], four reported a training phase [10,31,40,41], and two described both habituation and training [34,38]. Reporting the dimensions of the experimental set-up was an inclusion criterion for the selected studies. Considering the age of the animals, the studies included various age groups: four involved lambs (<6 months) [9,10,38,42], eight adults (6–20 months) [10,30,32,34,39,40,41,43], and four older animals (>2 years) [31,34,41,43]. Two studies used males only [41,42], four females only [30,31,32,39], and six both sexes [9,10,34,38,40,43].

#### 3.1.2. Sensory Discrimination and Association

Six manuscripts addressed sensory discrimination and association (Table 3). Four types of mazes and two arena-based tests have been proposed [22,23,31,42,44,45]. Regarding the pre-test period, two studies did not describe it [22,42], one reported a training phase [31], and three described both habituation and training [23,44,45]. Reporting the dimensions of the experimental set-up was an inclusion criterion for the selected studies. The studies included animals of various age: two involved lambs (<6 months) [22,42], two adults (6–20 months) [44,45], and two older animals (>2 years) [23,31]. One study used males only [42], two females only [31,44], and three both sexes [22,23,45].

#### 3.1.3. Problem-Solving and Decision-Making

Problem-solving and decision-making was addressed in six studies (Table 4). For this category, three types of maze tests and three different problem-solving task setups have been described [30,33,34,35,36,37]. Regarding the pre-test period, three studies reported a training phase [33,35,37], one included only habituation phase [30], and two described both habituation and training [34,36]. The studies included animals of various age groups: four involved adults (6–20 months) [30,33,34,37], and four older animals (>2 years) [34,35,36,37]. One study used males only [33], one females only [30], and four both sexes [34,35,36,37].

### 3.2. Type of Test

Of the 20 articles, 11 (55%) describe a maze [9,10,22,30,33,38,41,42,43,44,45], six describe a testing arena, referring to a rectangular setup [23,31,32,37,39,40], and three describe setups based on problem-solving objects: two use the cup task [35,36], one the bent tube [35], and one the cylinder task [34]. In the included sample of studies, Y-mazes, featuring obtuse angles, were more frequently described than T-mazes with right angles (five Y- [10,30,33,44,45] vs. three T- [22,38,42]). These papers describe different setups, characterized by varying dimensions, which can be summarized as follows: functional corridors (0.7–2.5 m width), confined arenas (e.g., 4.8 × 3.6 m to 8.5 × 2.5 m), and solid/opaque wall structures (1.0–1.8 m height).

Regarding the quadrilateral arenas the minimum dimensions are 4.5 × 3 m, while the maximum dimensions are 5.3 × 7 m and 8.7 × 3.1 m.

## 4. Discussion

### 4.1. Executive Cognitive Functions

#### 4.1.1. Cognitive Flexibility and Memory

Cognitive flexibility refers to an animal’s ability to adapt to different situations by applying strategies of varying complexity to achieve a reward. This also includes the ability to modify previously learned behaviors to adjust to environmental changes [52]. Cognitive flexibility is closely linked to cognitive memory, which enables both animals and humans to build the experience needed to navigate specific situations [53,54]. Untrained animals will generally respond less effectively compared to those already trained and equipped with specific memory tied to an event [55]. It is possible to identify these cognitive functions as among the main ones investigated, particularly from a translational perspective. This is reflected in the large number of studies and the considerable effort in the literature to understand, in greater detail, the processes underlying the ability to store experiences and to apply them adaptively and creatively in different contexts [30,39]. To evaluate these functions, the literature describes various methodologies that make use of pathways like mazes, larger arenas with objects and tools for interaction and cognitive association, and problem-solving tasks.

Mazes and labyrinths are described to be used in assessing an animal’s ability to navigate a familiar or novel path, eventually following periods of training and habituation, to reach a previously known goal. Successful task completion relies on spatial memory and cognitive flexibility.

The goal of using arenas is to evaluate the animals’ ability to associate a spatial path or to follow and recall a specific visual cue, often within a two-choice setup, in order to obtain a reward. In these cases, when training phases were included, they were particularly challenging, and habituation was almost always required (except in one instance). All arenas were rectangular.

Lastly, Knolle et al. [34] describe the use of cylinders to stimulate a cognitive and learning association. After a specific interval of 7 weeks, the memory capacity has been measured positively. Recall ability was not the primary objective of this manuscript.

#### 4.1.2. Sensory Discrimination and Association

Sensory discrimination and association involve the ability to differentiate between sensory cues and connect them to specific outcomes or rewards. These functions allow animals to interpret environmental stimuli, form associations, and adapt behaviors to optimize performance [56,57,58]. In sheep, such cognitive abilities are investigated to measure cognitive association, memory, and decision-making processes. As in other domestic mammals, the senses of sheep, such as vision, hearing, and smell, are highly developed, both for mother-lamb bonding and for species preservation, as prey animals. Among the type of test, both T-maze systems, integrated into more complex apparatuses [22,42], and modified Y-mazes are used [44,45]. Abecia et al. [22] and Morris et al. [45] studied hearing, while Bellegarde et al. [44] and Morris et al. [45] focused on vision and the recognition of images, colors, and faces, with an emphasis on association with a final reward. The described arenas, on the other hand, include two semi-automated two-choice operant systems for visual and acoustic association [23,31]. Abecia et al. [22] and Henrique et al. [42] avoid the pre-test period, incorporating stress as a variable to be analyzed in the study, while all other authors include a training and habituation period.

#### 4.1.3. Problem-Solving and Decision-Making

Problem-solving and decision-making represent essential executive cognitive functions that allow individuals to process information, make decisions, and adapt to new challenges. Problem-solving involves the ability to analyze a situation, self-control, recognize relationships, and generate strategies to achieve a goal [34]. Decision-making, on the other hand, reflects the capacity to acquire, learn, and apply knowledge based on experience, individual or social, enhancing future performance [59]. Together, these functions are crucial for understanding cognitive performance in sheep when faced with tasks that test spatial reasoning, decision-making accuracy, and inferential abilities. For this category, three types of maze tests and three different problem-solving task setups have been described. Regarding mazes: various approaches have been proposed, involving Y-mazes, or more generally, apparatuses that present the animal with the choice of proceeding to the right or left to reach a goal. Doyle et al. [33] and McBride et al. [37] emphasize the importance of specific training, while Morton and Avanzo [30] suggest a simple habituation period. McBride et al. [37], in addition to considering the complexity of the maze, introduce the concept of learning and association using mirrors. Problem-solving studies focus on deductive and inferential reasoning.

Regarding problem-solving tasks, three types have been described: cup task, tube task, and cylinder task. Duffrene et al. [35] tested sheep using cup and bent tube tasks in a familiar paddock, measuring their ability to solve increasingly complex tasks. Knolle et al. [34] further refined deductive reasoning tasks employing cylinder tasks to evaluate reasoning and reward retrieval after training with opaque or transparent objects. Nawroth et al. [36] used a sliding table with baited cups, adding control conditions to test cognitive reasoning. All these studies required a training period.

### 4.2. Type of Test

Our results showed that certain executive cognitive functions are closely linked to specific task designs. The choice largely depends on practical considerations such as available resources, space, and personnel. For example, the use of an arena requires only a relatively simple external structure and minimal staff involvement, whereas problem-solving stations involve multiple objects and require more intensive supervision. Therefore, there is no clear preference for one test over another, and they should ideally be used in combination to achieve a more comprehensive assessment of cognitive abilities [60,61]. Another key consideration is the temperament and personality of the animal: situations involving isolation or restraint should be avoided in animals that are easily stressed or not habituated, as these conditions can introduce clear biases in the results [62].

#### 4.2.1. Maze and Labyrinth

The most used apparatus in the selected papers is the Y-maze, to which structural modifications can be added. Other variants, such as the T-maze, the diamond-shaped maze, or spatial mazes, are also present. These labyrinths are particularly used in the study of cognitive functions, such as memory, cognitive flexibility, and associative learning, for example, concerning problem-solving abilities. The characteristic structure remains largely consistent: the walls are sufficiently high (taller than the animal) and are darkened (opaque, covered with black sheets, or made of materials that block external vision of the maze). These characteristics must ensure that the sheep can move around as comfortably as possible, avoiding freezing behaviors caused by the inability to turn around or having to turn back. In this regard, a habituation period is useful, whether in groups, in pairs, or with guidance, to minimize stress factors. Given that vision is one of the most developed senses in sheep, it is crucial to allow them to navigate in wide, comfortable spaces. Otherwise, confounding factors may arise. The selection of the apparatus depends not only on the specific cognitive function to be assessed but also on factors related to the species being studied. In the case of sheep, as with other prey animals, it is crucial to allow for greater freedom of movement.

Most of the literature highlights the importance of a reward-based test, such as food or the possibility of rejoining the flock or a companion. Therefore, a preliminary training phase is essential to ensure that the animal understands that a reward will be provided at the end of the test. In some studies, it has been observed that in sheep, a “win-shift” strategy is activated, especially when driven by appetitive motivation [38]. The habituation and training phases are generally omitted when the goal is to study the animal’s behavior in isolation, separated from the flock, and in animals with no prior experience with the apparatus, considered naïve animals, thus inexperienced.

#### 4.2.2. Testing Arena

Quadrilateral arenas are particularly well-described and widely used, both for the study of executive cognitive functions and for the analysis of cognitive emotionality, as well as behaviors related to stress, judgment bias, and isolation. Arenas can contain objects or, as in these cases, be empty (except for the presence of mirrors and screens in some setups [37]). Depending on the goal of the test, it is important to consider the level of isolation from the herd in individual tests, whether auditory or visual. Arenas with high walls, cameras positioned above or at the corners, final rewards, and automated test initiation systems (to minimize the confounding effects of human presence) form the foundation for this type of setup. This approach often includes a preliminary habituation and/or training phase, as mentioned in Section 4.2.1. The quadrilateral configuration is particularly useful for analyzing cognitive and sensory associations (e.g., reaching a target and discriminating based on images or sounds), in detour tests, or for assessing emotional factors such as fear, stress, or confidence.

#### 4.2.3. Problem-Solving Apparatus

Problem-solving devices are those that require the greatest number of objects to be used. These include movable gates, paths, and arenas with various types of objects, such as cylinders, cups, and tubes with different shapes and angles (right or flat). These tools represent the most complex method for analyzing executive cognitive functions, particularly the ability to solve tasks by integrating prior learning (training) with the capacity to modify or reformulate an adaptive response following changes to one or more characteristics of the initial task (reverse learning). These setups share the use of the animal’s experience, always including a training period to allow the animal to proceed and reach the reward. In two of the three studies, differences between species are also examined, particularly between sheep and goats. The animals are tested individually and subjected to repeated sessions to calculate averages and assess the reinforcement of specific knowledge. The arenas used for reward-seeking tests can vary in size: some are large and familiar (7 × 7 m) [35], while others are smaller, such as a paddock measuring 1.5 × 1.5 m [34]. In the latter case, a habituation period is required to reduce the animals’ stress, which could interfere with the feedback analyzed by researchers.

### 4.3. Limitations of This Scoping Review

Unlike systematic reviews, scoping reviews typically do not assess the quality of included studies, representing a notable weakness [63]. In our case, we did not evaluate the quality of cognitive tests, as our primary objective was to examine the available literature and compile an inventory of cognitive tasks previously described in sheep. Such definition and grouping of concepts constitute the foundation of an initial, exploratory review, aimed at identifying knowledge gaps and highlighting the variety of experimental apparatuses employed to achieve similar objectives. The search strategy was limited to studies published after 2010. This approach was designed to capture the most recent advances in executive cognition testing in sheep. Additionally, some of the inclusion criteria applied during the screening process were intentionally strict. For example, requirements regarding the presence of an abstract, the use of specific keywords, or the description of the test arena (e.g., size) were applied to ensure transparency and allow a consistent extraction of methodological details across studies. While these criteria helped standardize the dataset and facilitated comparison of experimental features, they may also have inadvertently excluded studies that assessed cognitive functions but did not report some methodological elements in sufficient detail. Similarly, studies evaluating cognition together with other behavioral outcomes may have been omitted when cognitive testing did not represent the primary focus of the article.

This outcome highlights the importance of precise terminology, accurate keyword selection, and clear titles in the original publications to ensure that relevant research can be effectively identified and included in future evidence syntheses. The review provides a structured overview of recent cognitive testing in sheep, while recognizing that some relevant studies may not have been captured due to the necessary boundaries imposed by the methodological approach.

## 5. Conclusions

In conclusion, the research revealed the presence of various tests used to assess some executive cognitive functions in sheep. This abundance highlights their extensive application in both translational research and, inevitably, livestock management. It was also observed that certain tests are employed to evaluate multiple cognitive functions, while others are more closely related to specific abilities. In particular, most studies examined flexibility and memory (13/20) and described spatial mazes (11/20); overall, the results support the notion that sheep are capable of performing complex cognitive tasks. The analysis of the articles suggests the importance of well-designed structures, which must be tailored to specific objectives. Environmental conditions should either encourage or limit exposure to certain stimuli, creating a context that promotes the sheep’s response. An appropriate reward is crucial for motivating the animal to complete the task, while proper pre-test preparation is essential when the goal is not solely to study the animal’s innate response to stimuli. Most of the tests highlighted specific attention to the design and functioning of the apparatuses, as well as the importance of habituation and training. These steps are necessary to prevent the sheep, a gregarious animal, from experiencing discomfort or stress due to isolation, which could alter behavioral responses. Future investigations should focus on the goal of standardizing testing systems, thus advancing knowledge about the sheep species and its cognitive processes, both physiological and pathological [64,65,66]. This could particularly support the application of such tests in the study of aging and neurodegenerative diseases affecting both animals and humans [31,67]. A better understanding of these processes could lead to benefits not only for livestock management but also for biomedical research and comparative medicine.

## Figures and Tables

**Figure 1 animals-15-03647-f001:**
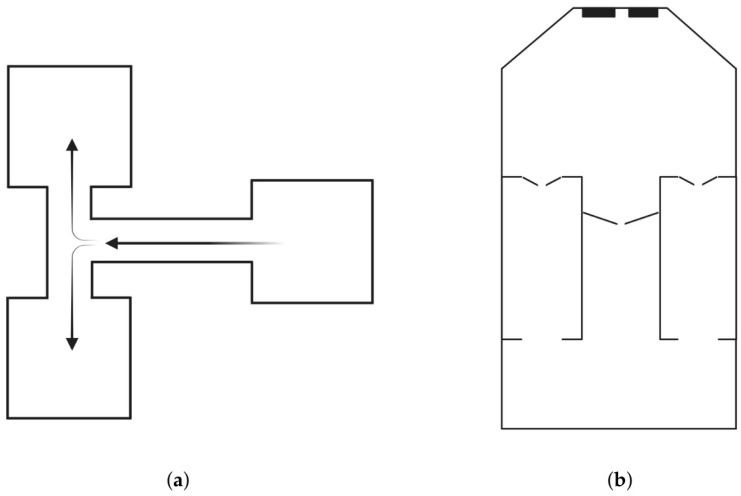
Schematic representation of two different typologies of apparatus: (**a**) a T-maze used by Abecia et al. [22]; (**b**) a mobile semi-automated system arena used by McBride et al. [23].

**Figure 2 animals-15-03647-f002:**
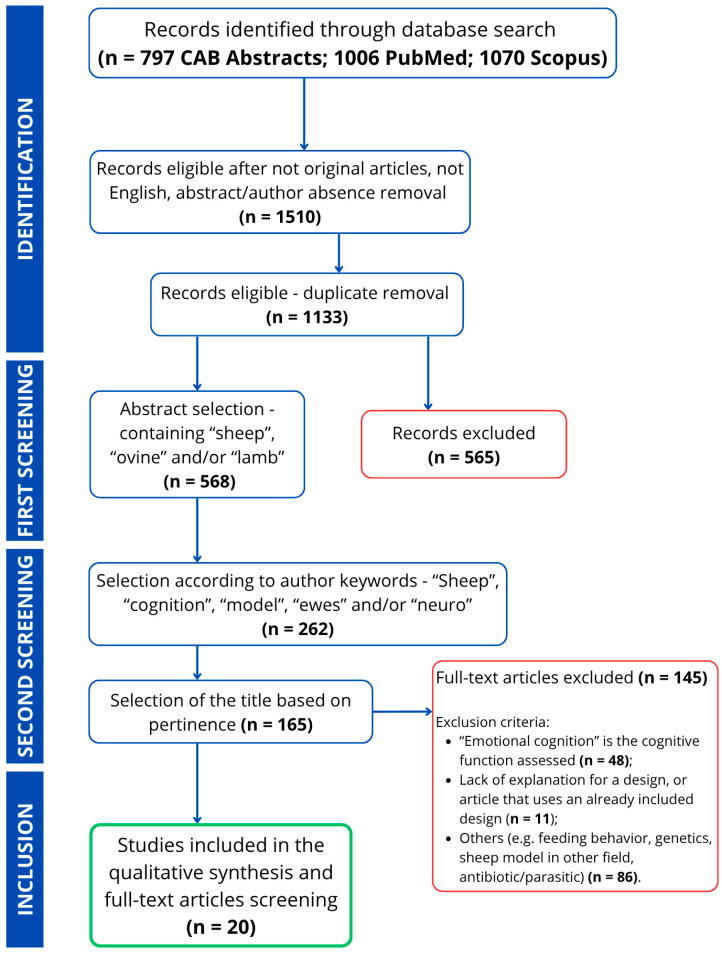
Modified PRISMA flow diagram with all the information on the number of excluded papers and the reason for exclusion, and inclusion.

**Table 1 animals-15-03647-t001:** Identification and description of articles divided according to the executive cognitive function investigated, with the addition of information related to the type of test used and the category and age to which the animals belong.

Cognitive Function	Typology of the Test	Animal Included (Category)	Sex	Age	Reference
**Cognitive flexibility**	Modified Y-maze	Prepubertal lambs and adult sheep	Both	18- and 40-week-old	[10]
	8 consecutive Y-mazes	Adult ewes	Female	~1 year-old	[30]
	Semi-automated for a two-choice task	Adult ewes	Female	7–8-year-old	[31]
	Spatial A-not-B detour task	Adult ewes and *adult goats (mixed sex)*	Female	~6–8-month-old	[32]
**Decision-making and problem-solving**	Spatial maze (modified Y-maze)	Castrated adult sheep	Male	6-month-old	[33]
	Cylinder task	“Old” and “young” sheep	Both	8-month-old and ~ 8-year-old	[34]
	Cup task and tube task	Adult sheep and *goats*	Both	2–10-year-old	[35]
	Cup task	Adult sheep and *goats (female)*	Both	~2-year-old	[36]
	8 consecutive Y-mazes	Adult ewes	Female	~1 year-old	[30]
	Locomotor-based decision-making task	Adult sheep	Both	~18–56-month-old	[37]
**Memory and spatial** **orientation**	Modified Y-maze	Prepubertal lambs and adult sheep	Both	18- and 40-week-old	[10]
	T-maze	Lambs	Both	2–5-month-old	[38]
	Two-choice discrimination task	Adult ewes	Female	17- and 19-month-old	[39]
	Spatial A-not-B detour task	Adult ewes and *adult goats (mixed sex)*	Female	~6–8-month-old	[32]
	Cylinder task	“Old” and “young” sheep	Both	8-month-old and ~ 8-year-old	[34]
	Food acquisition task in the arena	Yearling sheep	Both	From 10- and 30-week-old to 12-month-old	[40]
	Spatial maze	Lambs	Both	~1-month-old	[9]
	3 Spatial mazes	Adult sheep	Male	From 8-month-old to 2-year-old	[41]
	T-maze	Lambs	Male	30-day-old	[42]
	Spatial maze	Lambs and adult sheep (twins)	Both	8- 28- and 48-month-old	[43]
**Sensory discrimination and association**	T-maze	Lambs	Male	30-day-old	[42]
	T-maze, with isolation and a novel object	Lambs	Both	30-day-old	[22]
	Two-choice discrimination task (Y-maze)	Adult ewes	Female	10–12-month-old	[44]
	Semi-automated for a two-choice task	Adult ewes and castrated males	Both	37- and 25-month-old	[23]
	Y-maze	Adult sheep	Both	1-year-old	[45]
	Semi-automated for a two-choice task	Adult ewes	Female	7–8-year-old	[31]

**Table 2 animals-15-03647-t002:** Subdivision of the articles with memory and cognitive flexibility as the main topic under analysis, according to the specific choice of the type of test used and subsequent additional descriptions (pre-test period, dimension and architecture; timing and functioning).

Type of Test	Pre-Test Period	Dimension	Architecture	Timing and Functioning	Reference
**Maze**	Training (1 day)	1.5 m long arms.	Diamond-shaped maze, darkened walls, and a non-opaque ending gate.	Five days: after the training phase, tasks on learning, memory, and reversal were conducted.	[10]
	Habituation (total of 4 months)	8.5 × 2.5 m	8 rectangular modified Y-mazes, with corridors and gates following one another.	21 days: daily discrimination tasks (from 1 to 6 sets per day); correct choices were rewarded with food.	[30]
	Habituation (3 × 5 days), and 2 Training (exp. 1–5)	Start box (0.91 × 0.91 m), corridor (1.82 × 0.91 m), two goal arms (1.22 × 0.91 m).	A plywood T with a start box, a corridor, and two goal arms with hidden food.	5 exp.—nonmatching-to-place (reward in the opposite arm from the forced-run), matching-to-place (reward in the same arm as the forced-run), position habit (fixed rewarded arm on every trial), position habit reversal (rewarded arm reversed).	[38]
	Habituation (2 days)	7.5 × 3 m	Rectangular metal structure, movable maze barriers covered by opaque canvas.	2 trials during the same day: morning and afternoon. Behavioral aspects and completion time to rejoin the familiar group were recorded.	[9]
	Training (2 days)	10.8 × 4.6 m	Rectangular arena with maze barriers in different positions: Layouts 1, 2, and 3.	Weeks 41 and 83: spatial orientation tested in Layouts 1, 2; week 99: Layout 2 for long-term memory, Layout 3 for novel maze assessment.	[41]
	No	7.5 m long × 1.0 m wide × 1.20 m high and two arms (1.30 × 1.20 × 0.80 m each) at the end.	The maze has metal frames covered with black plastic and two arms.	3 days: lambs navigated the T-maze toward the maternal call; choice side, completion time, and predefined behavioral activities were recorded for each trial.	[42]
	No	Maximum of ~8 × 4.6 m	Rectangular arena with four main spaces, divided by metal walls.	3 trials in 3 time points: orientation and flexibility. Time taken to rejoin the familiar group was recorded.	[43]
**Arena**	Training (3 stages)	8.7 × 3.1 m and a final wall 2.2 m high.	Rectangular arena and a final wall.	Stop-signal task using both colors and sounds. 25 sessions: correct responses were rewarded; movements, reaction times, and stopping ability assessed.	[31]
	Habituation (familiar location with familiar people)	5.3 × 7 m	Large rectangular pen 5.3 × 7 m with a straight central barrier and a visible gap.	During the test, animals navigated a visible gap in a central barrier: four times in one position (A), then switched to another position (B) four times.	[32]
	No	2.2 × 6.6 m	Rectangular arena.	Sheep performed multiple (up to 100 trials) two-choice discrimination tasks; additional tests included reversal learning and spatial exploration tasks, with resting-state activity recorded.	[39]
	Training (18 individual runs)	1 m high plastic walls and gates.	Big rectangular arena divided into different spaces.	In the test phase, eight 1 min runs assessed each sheep’s food preferences.	[40]
**Problem-solving**	Habituation (2 × 5 min sessions) and Training (up to 7 sessions, 10 trials)	The cylinder task was carried out in a 1.5 × 1.5 m arena [23]	Rectangular testing area [23].	After seven weeks of training, sheep retrieved rewards from opaque or transparent cylinders.	[34]

**Table 3 animals-15-03647-t003:** Subdivision of the articles with sensory discrimination and association as the main topic under analysis, according to the specific choice of the type of test used and subsequent additional descriptions (pre-test period, dimension and architecture; timing and functioning).

Type of Test	Pre-Test Period	Dimension	Architecture	Timing and Functioning	Reference
**Maze**	No	7.5 m long × 1.0 m wide × 1.20 m high and two arms (1.30 × 1.20 × 0.80 m each) at the end.	The maze has metal frames covered with black plastic and two arms.	3 days: lambs navigated the T-maze toward the maternal call; choice side, completion time, and predefined behavioral activities were recorded for each trial.	[42]
	No	Chamber (2 × 2 m), corridor (4 × 0.70 m)	Three different chambers connected as a T-maze through a corridor	After 20 s in the isolation pen, lambs were individually tested (5 min) using sound cues.	[22]
	Habituation (2 days) and training (around 20 sessions)	4.8 × 3.6 m	Inside the global arena a two-armed maze is built, following a decision area equipped with screens displaying cues.	First training on colored cards; second phase on facial cues. Finally, two test sessions were conducted to generalize the learned associations to new and familiar faces.	[44]
	Habituation (5 weeks) and training (5 days)	Arena (2 × 2.85 m) Central solid wall (1.8 m high)	A Y-maze inside a sheep pen with solid wood separating the two feeding buckets.	During testing, sheep made choices in the Y-maze, guided by visual or auditory cues that led to food rewards (4 trials per day over 10 days).	[45]
**Arena**	Habituation (4 × 15 min sessions), and Training (10 trials each, ~ 4–8 min)	8.7 × 3.1 m and a final wall 2.2 m high.	Rectangular arena and a final wall.	Visual discrimination and memory task (up to 13 sessions). Correct responses earned rewards. Learning criteria were identified.	[23]
	Training (3 stages)	8.7 × 3.1 m and a final wall 2.2 m high.	Rectangular arena and a final wall.	Stop-signal task using both colors and sounds. 25 sessions: correct responses were rewarded; movements, reaction times, and stopping ability assessed.	[31]

**Table 4 animals-15-03647-t004:** Subdivision of the articles with problem-solving and decision-making as the main topic under analysis, according to the specific choice of the type of test used and subsequent additional descriptions (pre-test period, dimension, and architecture; timing and functioning).

Type of Test	Pre-Test Period	Dimension	Architecture	Timing and Functioning	Reference
**Maze**	Training (5 days)	20 × 9 m	Rectangular arena with two error zones, corridors, and opaque exterior walls.	Three days: they were pre-exposed to food reward or a dog. Afterward, they ran the maze to assess stress and distraction effects on cognitive performance.	[33]
	Training (8.3 ± 3.2 sessions (10 trials each))	Training mirror exposure: two arenas (3 × 4 m) and (5 × 15 m), with different mirror positions.	Training spatial: rectangular maze with gates, grates, and a visible bucket. Testing: same as training spatial, with mirrors.	Mirror exposure followed the training, with behavioral analysis during sessions (10 trials) to assess decision-making accuracy.	[37]
	Habituation (total of 4 months)	8.5 × 2.5 m	8 rectangular modified Y-mazes, with corridors and gates following one another.	21 days: daily discrimination tasks (from 1 to 6 sets per day); correct choices were rewarded with food.	[30]
**Problem-solving**	Training (3 phases)	7 × 7 m 1.20 m high	The tests were conducted in a familiar paddock, covered with a roof and with wooden board fences.	Cup task: 10 sessions with 8 trials each to test deductive reasoning. Bent tube task: 12 sessions with 10 trials each, testing inferential ability.	[35]
	Habituation (2 × 5 min sessions) and Training (up to 7 sessions, 10 trials)	The cylinder task was carried out in a 1.5 × 1.5 m arena [23]	Rectangular testing area [23].	After seven weeks of training, sheep retrieved rewards from opaque or transparent cylinders.	[34]
	Habituation (at least 4 trials) and training (at least 2 sessions and 10 trials)	1.2 × 2.7 m 60 × 25 cm sliding table is used for the cups.	Sheep were kept in a pen, visually isolated but with sensory contact.	Ex 1: 10 sessions with 8 trials, testing 4 conditions (both, baited, empty, control) and 5 s cues. Bowls were randomly baited. Ex 2: like Ex 1, with inner cups to prevent local enhancement	[36]

## Data Availability

No new data were created or analyzed in this study. Data sharing is not applicable.

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
