# Peer review of "Assessing Executive Cognitive Functions in Sheep: A Scoping Review of Recent Literature on Cognitive Tasks"

_animals, 2025, doi:10.3390/ani15243647_

Round 1

Reviewer 1 Report

Comments and Suggestions for Authors

Galanti et al have addressed an interesting question, seeking to identify and collate studies of cognitive function in the sheep. This work has the potential to be of value, given the importance of this species for studies relevant to both animal welfare and human clinical outcomes, such as understanding changes following brain injury or impacts of developmental exposures. Unfortunately, the authors have not adequately searched the literature to identify relevant studies, the selection of included articles is not clearly defined, and the summaries of available literature would be improved by providing more detail. We hope that the feedback below can be used to improve subsequent studies. We strongly recommend that if the authors wish to conduct a rigorous scoping review of this question that they undertake a training course to gain the required skills, such as those offered by JBI.

Major issues:

  1. Justification of question: Much of the introduction is not clearly relevant to the proposed question, eg. lines 63-68 defining health. It is not clear why the detailed description of multiple tests used in rodents (lines 73-90) is included – how this relates to the need to evaluate sheep cognition and the tests used to study this is unclear.

  1. Lines 115-117: Conducting a scoping review does not necessarily lead to conducting a systematic review, and is not a prerequisite to a systematic review, suggesting a lack of understanding of the purpose of this review type. The scoping review identifies existing literature in order to identify knowns and unknowns and strengths and weaknesses of available literature, whereas the systematic review compiles and synthesises evidence to evaluate the impact of a factor on a specific outcome or set of outcomes.

  1. Search approach:
    1. The adherence to the PRISMA guidelines is adequately described in the methodology, however the eligibility criteria described on lines 156-161 are weakly justified.
    2. Inclusion period: Restricting to studies conducted after 2010 omits key studies from the early 2000s that helped shape the field (e.g. Kendrick et al. on face recognition in a discrimination task). While the authors do acknowledge the existence of foundational studies, no references are provided.
    3. Search string: The search string will not identify all relevant literature – key terms/MeSH terms should be used in addition to wild-card characters. For example the current search will not include studies where “cognition” rather than “cognitive” is in the article. It is also not clear which fields in each database have been searched. Is this title-abstract-keywords?
    4. Search scope: Furthermore, use of only two databases has likely contributed to omission of relevant literature, particularly studies where sheep have been used in studies of clinically-relevant questions. At a minimum, searching PubMed and Embase would be required for the initial searching of literature. The current review excludes important studies, including recent studies as well as those published before 2010, e.g.:
      • Hernandez et al 2009 Behav Brain Res PMID: 19467268
      • Cock et al 2001 Clin Exp Pharmacol Physiol PMID: 1173400
      • Chasles et al 2024 Neuroendocrinology PMID: 38697024

  1. Study screening, inclusion and exclusion: We have major concerns about all stages of the screening process used in this scoping review. Clear inclusion and exclusion criteria have not been provided. Importantly, the screening process used was inappropriate and is not clearly described. Based on the PRISMA diagram, articles were excluded if the abstract was missing – this is not appropriate as you do not have sufficient information to make this decision – instead this source should be taken through to full-text screening for evaluation. The selection based on inclusion of one of four keywords is similarly inappropriate and will not include all relevant studies in which sheep cognition is evaluated. Screening titles only means that articles where these outcomes are included in abstracts will not be included (see PRISMA diagram “selection of the title based on pertinence!”). Selecting articles based on cognitive outcomes being stated as the main objective of the study (lines 183-184) fails to capture studies in which cognitive outcomes are part of a suite of outcomes relevant to the experimental question and inappropriately excludes important studies and test approaches. Importantly, good practice in a scoping or systematic review requires dual independent screening and then decisions on exclusion/inclusion with a third party by consensus. There is substantial risk of bias since the screening process was carried out by only one of the authors. It is best practice for scoping review screening to be carried out by at least two reviewers. Use of an online tool such as Covidence facilitates this process.

  1. Categorisation of articles into domains: This approach seems rather arbitrary and not helpful inn separating groups of test. In particular, most tests of memory first require a learning phase. Furthermore, at least some studies are categorised incorrectly. For example, the two-choice tests described in ref 18 are based on visual discrimination, which seems to fit into Table 3?

  1. Table 1 categorises the reviewed studies into domains defined in the manuscript. This table lacks sufficient detail given its purpose to provide descriptive information from each study. Animal age is referred to in the table as “adult sheep” or “old and young sheep”, however extracting the numerical age of the sheep used in each study (where possible) would better facilitate reproducibility for future studies. Why are goats included in this summary of outcomes in sheep? If sex is an important characteristic it should be provided for all studies. If the characteristics of animals are considered important, then it seems contradictory for the authors to have excluded subsequent studies in different animal groups on the basis that they used the same apparatus!

  1. Discussion of tests: Maze types and dimensions are tabulated by the defined domains and discussed in the sections to follow; however, this discussion does not include critical analysis of the data presented in the tables. There is minimal interpretation of how differences in spatial layout, motivation, or social isolation affect task performance in sheep. While the authors state that the “primary objective was to examine the available literature and compile an inventory of cognitive tasks previously described in sheep”, experimental validity, reliability and outcome measures are not discussed – all central to assessing the suitability of a task as a cognitive assessment. Discussion of these elements would better facilitate the compilation of an inventory of cognitive tasks that could guide selection of appropriate tests for future studies of sheep cognition. Lines 225-230 are extremely unclear. The evaluations of tests and approaches throughout seems subjective.

  1. Section 3.3: It is not clear that this is relevant as the points raised are not within the scope of this review and do not cite literature identified in this review.

  1. Standardisation as a recommendation: Standardisation improves comparability not reliability (lines 40-41). Importantly, we disagree with the authors recommendation that tests should be standardised (conclusion and throughout). Standardising tests is not always appropriate, depending on the question of specific studies, since different tests are best used to assess different cognitive abilities. Furthermore, requirements and timelines for training may mean that some gold standard tests cannot be used in specific experimental situations, such as testing behavioural outcomes in neonatal animals. Availability of test apparatus, time and resources for conducting testing may also preclude use of specific tests. Indeed, as made clear from the introduction of rodent tests, multiple tests are also used in other species. Other discussion and recommendations such as the need to include and describe training and acclimatisation are more useful to the field.

Minor points for revision:

  1. Introduction, lines 105-108: This does not reflect the importance of preclinical models of human conditions in sheep, conducted in order to understand the disease/effects of an exposure, not for the purpose of comparisons between species. These ideas are more clearly captured in the abstract.

  1. Figure 1: Why is this included in the introduction? The relevance is unclear, as is the basis for selecting these two examples of testing apparatus. It is not clear why test shapes rather than different test types are the focus here. As these are published studies, not just plans for future studies, “proposed” should be replaced by “used” in the Figure 1 title.

  1. Table 1, first line: prepubertal not prepuberal

  1. Results & discussion, lines 198-201 –There is inconsistency in the way cognitive definitions are referenced which mars the clarity of the paper. For example, the third category is originally defined as “decision making and problem solving” but is later referred to as “problem solving and learning” (line 213). What is meant by “grouped, from the authors”? Which authors – scoping review authors or based on statements in sources?

  1. Lines 201-202: Should this read “Tables 2-4” at the start of line 202?

  1. Lines 242-243: This statement is not correct relative to the tests cited – for example ref 18 uses tests of image discrimination with the side on which the correct image is displayed randomised – it does not test the ability to learn a spatial path.

  1. Lines 323-326 and 352-355: Conclusions about best design based on inclusion of the study in this review are problematic, since the review has not adequately search or included relevant literature. Shape of the arena (square vs rectangular) will not impact the ability of sheep to detour around an obstacle, for example.

  1. Lines 414-416: It is unclear how citation rates allow you to reach this conclusion!

  1. References: Many references in the list are incomplete, lacking volume, article and page numbers.

Author Response

We sincerely thank Reviewer 1 for the thoughtful feedback, especially for the constructive suggestions. All the revisions made in response to the comments are detailed below. We believe these suggestions have substantially improved the manuscript.

Please refer to the lines in the revised manuscript with track changes.

Major issues

Comments 1: Justification of question: Much of the introduction is not clearly relevant to the proposed question, eg. lines 63-68 defining health. It is not clear why the detailed description of multiple tests used in rodents (lines 73-90) is included – how this relates to the need to evaluate sheep cognition and the tests used to study this is unclear.

Response 1: We thank the reviewer for this comment. We have revised the introduction to improve the logical flow, clarifying the relevance of each section. We also updated the Abstract. The reference to health was meant to emphasize its close interconnection with welfare and cognition. We have rephrased this part to make the relationship more concise and aligned with the focus of the review. See Lines 106-111. Regarding rodent studies, our intention was only to provide brief contextual background on the most commonly used cognitive tests in translational research. We have clarified this point in the revised text, highlighting that sheep may offer complementary advantages due to their brain structure and lifespan. We have added the following clarifying sentence to highlight this rationale. See Lines 114-116.

Comments 2: Lines 115-117: Conducting a scoping review does not necessarily lead to conducting a systematic review, and is not a prerequisite to a systematic review, suggesting a lack of understanding of the purpose of this review type. The scoping review identifies existing literature in order to identify knowns and unknowns and strengths and weaknesses of available literature, whereas the systematic review compiles and synthesises evidence to evaluate the impact of a factor on a specific outcome or set of outcomes.

Response 2: To address this point, we have revised the text as follows: “Conducting a systematic review to address a practical problem represents the highest level of evidence in the hierarchy of evidence-based medicine. When properly executed, a systematic review offers a rigorous and replicable method for identifying, evaluating, and summarizing existing evidence. A scoping review is a distinct but equally valuable form of knowledge synthesis. It is typically conducted to map and summarize the available literature on a given topic, especially when a systematic review is not feasible; such as when the evidence base includes studies with diverse designs, methodologies, or objectives”. See Lines 173-191.

Comments 3: Search approach:

The adherence to the PRISMA guidelines is adequately described in the methodology, however the eligibility criteria described on lines 156-161 are weakly justified. Inclusion period: Restricting to studies conducted after 2010 omits key studies from the early 2000s that helped shape the field (e.g. Kendrick et al. on face recognition in a discrimination task). While the authors do acknowledge the existence of foundational studies, no references are provided.

Search string: The search string will not identify all relevant literature – key terms/MeSH terms should be used in addition to wild-card characters. For example the current search will not include studies where “cognition” rather than “cognitive” is in the article. It is also not clear which fields in each database have been searched. Is this title-abstract-keywords?

Search scope: Furthermore, use of only two databases has likely contributed to omission of relevant literature, particularly studies where sheep have been used in studies of clinically-relevant questions. At a minimum, searching PubMed and Embase would be required for the initial searching of literature. The current review excludes important studies, including recent studies as well as those published before 2010, e.g.: Hernandez et al 2009 Behav Brain Res PMID: 19467268

Cock et al 2001 Clin Exp Pharmacol Physiol PMID: 1173400

Chasles et al 2024 Neuroendocrinology PMID: 38697024

Response 3: We are deeply grateful to the reviewer for this thorough and constructive feedback, which we found extremely valuable in refining both the current manuscript and our overall research approach. Regarding the eligibility criteria, our main intention was to focus specifically on Executive Cognitive Functions, while excluding studies primarily centered on Emotional Cognition. The goal was to outline the strengths and gaps in the literature addressing executive functioning, with particular emphasis on the design and replicability of tasks involving arenas, mazes, and problem-solving paradigms. During the second-level screening, we carefully examined the full texts (See Materials and Methods section) to verify whether schematic representations of experimental setups were provided. To avoid redundancy, tasks reported by the same research group using identical animals and apparatus were included only once (e.g., Hunter et al., 2015; PMID: 25449405 and PMID: 27759494). On the inclusion period, we intentionally limited the time frame to studies published after 2010 to emphasize the most recent developments and the growing scientific interest in this area. Foundational studies from the early 2000s were acknowledged through their citation in more recent research; for instance, Hernandez et al. (2009; PMID: 19467268) and Erhard et al. (2004; PMID: 15084418) have been used by Hunter et al. (2015; PMID: 25449405), while Marin et al. (1997; DOI: 10.1016/S0168-1591(97)00071-3) has been used by Abecia et al. (2014; PMID: 24847694); and many others. Thus, earlier studies were considered indirectly through their integration into newer, methodologically updated works.

Regarding the search string, we completely agree with the reviewer’s point. We have refined the search strategy by confirming that searches were performed within the title, abstract, and keyword fields (See Line 217), and by incorporating broader wildcard terms (e.g., “cogniti*”) to ensure that all relevant variations such as “cognition” and “cognitive” were captured (See Lines 218-219). We sincerely thank the reviewer for highlighting this important refinement, which helped make the search more comprehensive. As for the search scope, we initially selected Scopus and CAB Abstracts for their wide coverage of animal cognition research. However, following the reviewer’s valuable advice, we expanded our search to include PubMed as a third database (See Lines 209-210). Although this significantly increased the number of screened papers, the final inclusion process, applying the same criteria as before, resulted in the same set of 20 studies, most of which were retrieved across all three databases. See both M&M and Results sections.

Finally, regarding Chasles et al. (2024; PMID: 38697024), we confirm that this study was identified in the initial search but excluded during the title-screening stage because it explicitly focused on anxiety-like behavior, which we categorized under emotional cognition rather than executive functioning. We have clarified this rationale in the revised text to make the exclusion criteria more transparent.

We truly appreciate the reviewer’s insightful comments, which led us to revisit and strengthen key methodological aspects. We have carefully considered every suggestion and implemented all feasible improvements to enhance clarity, transparency, and rigor throughout the manuscript. We added also a more precise paragraph in Limitations section, explaining the importance of terminology. See Lines 1178-1185.

Comments 4: Study screening, inclusion and exclusion: We have major concerns about all stages of the screening process used in this scoping review. Clear inclusion and exclusion criteria have not been provided. Importantly, the screening process used was inappropriate and is not clearly described. Based on the PRISMA diagram, articles were excluded if the abstract was missing – this is not appropriate as you do not have sufficient information to make this decision – instead this source should be taken through to full-text screening for evaluation. The selection based on inclusion of one of four keywords is similarly inappropriate and will not include all relevant studies in which sheep cognition is evaluated. Screening titles only means that articles where these outcomes are included in abstracts will not be included (see PRISMA diagram “selection of the title based on pertinence!”). Selecting articles based on cognitive outcomes being stated as the main objective of the study (lines 183-184) fails to capture studies in which cognitive outcomes are part of a suite of outcomes relevant to the experimental question and inappropriately excludes important studies and test approaches. Importantly, good practice in a scoping or systematic review requires dual independent screening and then decisions on exclusion/inclusion with a third party by consensus. There is substantial risk of bias since the screening process was carried out by only one of the authors. It is best practice for scoping review screening to be carried out by at least two reviewers. Use of an online tool such as Covidence facilitates this process.

Response 4: We are grateful to the reviewer for highlighting this important issue. We have now expanded and clarified the description of the screening, inclusion, and exclusion process to ensure full transparency. The screening was conducted by the first author and independently verified by the second author, which is now explicitly stated in the manuscript. This step has now been explicitly mentioned in the manuscript to avoid any ambiguity (see also Author Contributions section). See Line 1268. We have also clarified that, whenever studies appeared ambiguous, full-text screening was carried out. These revisions make our selection process more transparent and aligned with best practices.

Comments 5: Categorisation of articles into domains: This approach seems rather arbitrary and not helpful inn separating groups of test. In particular, most tests of memory first require a learning phase. Furthermore, at least some studies are categorised incorrectly. For example, the two-choice tests described in ref 18 are based on visual discrimination, which seems to fit into Table 3?

Response 5: We thank the reviewer for this insightful comment. We fully acknowledge the conceptual overlap among the different cognitive domains assessed in the reviewed studies and agree that several tasks, including the two-choice tests described in McBride et al. (2016), could potentially fit into more than one category (See Lines 404-405). The three-domain framework was adopted simply as a practical way to present the data clearly, not as a definitive classification. Regarding the reviewer’s example, we agree that the cited task involves both visual discrimination and recall components. This duality precisely illustrates the methodological complexity we aimed to highlight. Many cognitive tests simultaneously engage multiple executive cognitive functions and therefore cannot be neatly assigned to a single category. See Table 1.

Comments 6: Table 1 categorises the reviewed studies into domains defined in the manuscript. This table lacks sufficient detail given its purpose to provide descriptive information from each study. Animal age is referred to in the table as “adult sheep” or “old and young sheep”, however extracting the numerical age of the sheep used in each study (where possible) would better facilitate reproducibility for future studies. Why are goats included in this summary of outcomes in sheep? If sex is an important characteristic it should be provided for all studies. If the characteristics of animals are considered important, then it seems contradictory for the authors to have excluded subsequent studies in different animal groups on the basis that they used the same apparatus!

Response 6: We thank the reviewer for these helpful comments. To improve the clarity and reproducibility of Table 1, we have added a new column titled “Age”, where the numerical age of the animals has been reported whenever it was available in the original studies (some studies actually referred to a range of ages rather than a specific time point). Regarding sex, this information was already included in the “Category” column and has now been checked and standardized across all entries to ensure consistency. As for the inclusion of goats, we acknowledge that they are not the main focus of this review. However, some of the studies included a direct comparison between sheep and goats. In those cases, we chose to briefly mention the presence of goats to give readers a complete understanding of the study design and context. Importantly, data from goats were not used in defining maze or arena types but only referenced when necessary to clarify experimental setups.

Comments 7: Discussion of tests: Maze types and dimensions are tabulated by the defined domains and discussed in the sections to follow; however, this discussion does not include critical analysis of the data presented in the tables. There is minimal interpretation of how differences in spatial layout, motivation, or social isolation affect task performance in sheep. While the authors state that the “primary objective was to examine the available literature and compile an inventory of cognitive tasks previously described in sheep”, experimental validity, reliability and outcome measures are not discussed – all central to assessing the suitability of a task as a cognitive assessment. Discussion of these elements would better facilitate the compilation of an inventory of cognitive tasks that could guide selection of appropriate tests for future studies of sheep cognition. Lines 225-230 are extremely unclear. The evaluations of tests and approaches throughout seems subjective.

Response 7: We sincerely thank the reviewer for this insightful comment, which helped us strengthen the discussion. We fully agree that a deeper analysis of validity, reliability, and outcome measures would add value; however, in accordance with the methodological framework for scoping reviews (Arksey & O’Malley, 2005), our primary aim was to map and organize existing literature rather than critically appraise each study. Nevertheless, following the reviewer’s suggestion, we have revised Section 4.1.1. (“Cognitive flexibility and memory”) to improve clarity and include a more explicit discussion of how differences in maze design, motivation, and social context may influence task performance. We also added a note highlighting the importance of future systematic assessments of validity and reliability across tasks. We are very grateful for this comment, which prompted meaningful improvements in focus and interpretative depth. See Lines 904-908 in the revised manuscript.

Comments 8: Section 3.3: It is not clear that this is relevant as the points raised are not within the scope of this review and do not cite literature identified in this review.

Response 8: We sincerely thank the reviewer for this comment. We fully understand that the content of Section “A Note on Personality” may appear beyond the main scope of the review, as it only indirectly relates to the literature analyzed. To avoid any possible confusion or misinterpretation, we have removed this section as suggested. We truly appreciate this remark, which helped us make the manuscript more focused and coherent.

Comments 9: Standardisation as a recommendation: Standardisation improves comparability not reliability (lines 40-41). Importantly, we disagree with the authors recommendation that tests should be standardised (conclusion and throughout). Standardising tests is not always appropriate, depending on the question of specific studies, since different tests are best used to assess different cognitive abilities. Furthermore, requirements and timelines for training may mean that some gold standard tests cannot be used in specific experimental situations, such as testing behavioural outcomes in neonatal animals. Availability of test apparatus, time and resources for conducting testing may also preclude use of specific tests. Indeed, as made clear from the introduction of rodent tests, multiple tests are also used in other species. Other discussion and recommendations such as the need to include and describe training and acclimatisation are more useful to the field.

Response 9: We fully acknowledge that complete standardization is not always appropriate or feasible, as the optimal choice of cognitive test depends on the specific research question, the target cognitive domain, and practical factors such as animal age, training requirements, equipment availability, and experimental timelines. Our intention was only to encourage greater methodological consistency and transparency, rather than to advocate for rigid uniformity. We have revised the text to clarify that our recommendation focuses on improving comparability and reliability across studies. Lines 38, 199. The aim is to encourage the use of more consistent and well-described procedures, particularly regarding habituation and training phases, to enhance the reproducibility and interpretability of findings across studies. We appreciate the reviewer’s note that these recommendations are useful, and we have adjusted the conclusion accordingly to ensure the message is balanced and accurately framed.

Minor points for revision

Comments 1: Introduction, lines 105-108: This does not reflect the importance of preclinical models of human conditions in sheep, conducted in order to understand the disease/effects of an exposure, not for the purpose of comparisons between species. These ideas are more clearly captured in the abstract.

Response 1: We thank the reviewer for this valuable and thoughtful comment. We agree that the sentence in the Introduction did not adequately reflect the importance of using sheep as preclinical models of human conditions and could be misinterpreted as emphasizing comparative cognition between species. Our intention is to highlight the increasing relevance of sheep in translational and biomedical research, where they are used to study mechanisms of neurodegenerative diseases, aging, and the effects of specific exposures. These models are crucial for understanding human pathophysiology and cognitive processes in a controlled, ethically responsible framework, rather than for direct interspecies comparisons. To align the Introduction more closely with the ideas expressed in the Abstract and to accurately represent the purpose of these studies, we have revised the text. See Lines 162-164 and the Abstract. This revision now clarifies the translational and preclinical focus of research on sheep cognition, ensuring consistency between the Abstract and Introduction and better reflecting the intent of the review.

Comments 2: Figure 1: Why is this included in the introduction? The relevance is unclear, as is the basis for selecting these two examples of testing apparatus. It is not clear why test shapes rather than different test types are the focus here. As these are published studies, not just plans for future studies, “proposed” should be replaced by “used” in the Figure 1 title.

Response 2: We included a representative illustration of two different testing schemes to assist readers in visualizing the types of apparatus described later in the manuscript. These two examples were selected from the 20 papers included because (1) they are simpler to illustrate, given their less complex architecture, and (2) they are among the most commonly used and frequently cited in the literature. We believe this visual representation helps clarify the diversity of approaches discussed in the text. Following the reviewer’s suggestion, we have also replaced the word “proposed” with “used” in the Figure 1 title. See Lines 168-170.

Comments 3: Table 1, first line: prepubertal not prepuberal

Response 3: We changed “prepuberal” to “prepubertal” in Table 1.

Comments 4: Results & discussion, lines 198-201 –There is inconsistency in the way cognitive definitions are referenced which mars the clarity of the paper. For example, the third category is originally defined as “decision making and problem solving” but is later referred to as “problem solving and learning” (line 213). What is meant by “grouped, from the authors”? Which authors – scoping review authors or based on statements in sources?

Response 4: We agree. As also discussed in the Discussion section, it is indeed difficult to assign a single test to a single cognitive function, and vice versa. Our intention was to provide a simplified and reader-friendly framework to classify the various cognitive functions, using tables to help readers follow the logical flow. We do not aim to propose a definitive classification; rather, given the absence of a clear and established framework, particularly for sheep, we sought to present a practical way to organize the information. Regarding the third category, “problem-solving and decision-making,” we acknowledge that the mention of “learning” was an oversight. This has now been corrected. We have also clarified the wording to specify that the categories were grouped by the authors of the present paper. See Lines 401-403 and 733.

Comments 5: Lines 201-202: Should this read “Tables 2-4” at the start of line 202?

Response 5: We thank the reviewer for noticing this formatting issue. Yes, it should read “Tables 2, 3, and 4,” and this has been corrected in the revised manuscript. See Line 405.

Comments 6: Lines 242-243: This statement is not correct relative to the tests cited – for example ref 18 uses tests of image discrimination with the side on which the correct image is displayed randomised – it does not test the ability to learn a spatial path.

Response 6: We thank the reviewer for this clarification. We agree that the previous wording could be misleading, as the cited reference (McBride et al., 2016) does not directly assess the ability to learn a spatial path but rather focuses on image discrimination and recognition processes. For this reason, we have corrected the sentence to better reflect the type of test described and its relevance to memory assessment. McBride et al. was included under the “memory” category because the task still involves recall components, which are considered part of the broader cognitive domain of memory. See Line 916-918.

Comments 7: Lines 323-326 and 352-355: Conclusions about best design based on inclusion of the study in this review are problematic, since the review has not adequately search or included relevant literature. Shape of the arena (square vs rectangular) will not impact the ability of sheep to detour around an obstacle, for example.

Response 7: We agree with the reviewer’s observation. For this reason, we have partially revised the conclusion of Section 3.2. Type of Test, which now reads: “In this review, Y-mazes, featuring obtuse angles, were more frequently described than T-mazes with right angles (five Y vs. three T).” We chose to maintain a purely descriptive comment based on our findings, avoiding any subjective interpretations (e.g., using terms such as “prefer”). See Lines 742-745. Regarding the second point, we thank the reviewer for noting that only rectangular arenas were reported in the included studies. Our intention was not to imply that rectangular arenas are the only type used, but simply that they were the most frequently described. During the full-text analysis, we also identified some triangular and square arenas, although these were employed for assessing emotional or social cognition (e.g., mother–offspring bonding or human–animal interaction). To prevent possible misinterpretation, we have replaced the term “rectangular arenas” with “four-lateral arenas”. See Line 1099 and 1109.

Comments 8: Lines 414-416: It is unclear how citation rates allow you to reach this conclusion!

Response 8: The conclusion was not based solely on citation rates but rather on the analysis of the full text of the 20 articles included in this review. The vast majority clearly referenced one or more previous articles that they drew upon to design their “own” setup. To clarify this point, we have modified the sentence and added in the Results section. See Lines 409-411.

Comments 9: References: Many references in the list are incomplete, lacking volume, article and page numbers.

Response 9: We thank the reviewer for bringing this to our attention. We manually checked the entire reference list and identified some citations that were missing page numbers and/or volume information, likely due to an issue related to the use of Mendeley. All missing details have now been added, and the reference list has been fully updated.

Reviewer 2 Report

Comments and Suggestions for Authors

Dear authors and editor, the work is well done; it is interesting and innovative, but some aspects need to be corrected:

Materials and methods: 
Why weren't other databases used? WOS, for example? 

Tables 2, 3, and 4 lack a uniform format and become very difficult to read. It would be helpful to add more columns to make the table easier to read (e.g., dimension column, architecture column, etc.).

There is no clear separation between the presentation of results and the discussion. The report tends to be very descriptive and does not contrast which tests would be the best or worst (although it is described within the limitations). Without this discussion, the report is descriptive. 

Some other comments 
Lines 83-95: Add bibliographic support for these ideas 
Lines 111-112: Adjust font format to what is required 
Line 151: and/or?
Line 216: There is a lack of discussion about the advantages and disadvantages of the most commonly used methods for measuring cognitive flexibility and memory. 
Line 222: Change “:” to “.”

Author Response

We would like to thank Reviewer 2 for their valuable comments and suggestions. Below, we present all the revisions made in response to these comments. We believe that incorporating these suggestions has resulted in a clearer and more comprehensive manuscript.

Please refer to the lines in the revised manuscript with track changes.

Comments 1: Materials and methods: Why weren't other databases used? WOS, for example? 

Response 1: We thank the reviewer for this valuable suggestion. In our study, we initially focused on Scopus and CAB Abstracts, as they were considered appropriate and sufficiently comprehensive for addressing our research objectives. However, following the reviewer’s advice, we also included PubMed as a third database. Although this expanded the number of papers screened, after applying the same selection criteria, we ultimately obtained the same 20 studies to include, most of which were retrieved across the three databases. See Materials and Methods section.

Comments 2: Tables 2, 3, and 4 lack a uniform format and become very difficult to read. It would be helpful to add more columns to make the table easier to read (e.g., dimension column, architecture column, etc.).

Response 2: We thank the reviewer for this helpful comment. Following the suggestion, we have revised the tables by splitting the previous “dimension and architecture” column into two separate columns to improve clarity and readability. However, it was not possible to further divide the “timing and functioning” column, as not all the papers included provided complete information on the functioning design, making systematic comparison difficult. See Table 2, 3 and 4.

Comments 3: There is no clear separation between the presentation of results and the discussion. The report tends to be very descriptive and does not contrast which tests would be the best or worst (although it is described within the limitations). Without this discussion, the report is descriptive. 

Response 3: To improve the separation between the Results and Discussion sections, we have created two distinct chapters. The first chapter presents the Results, including the findings from the literature search, the tables with initial categorization, and the subsequent division according to tasks and the executive cognitive functions assessed. The Discussion begins after Table 4 and provides descriptive commentary on the tables above. No new information was added or modified in the revised manuscript; some content was simply relocated from one section to another. The aim of these comments is not to rank methods as best or worst, but rather to compile a comprehensive catalogue and provide an inventory of approaches, as noted in the limitations. Additionally, in response to Comment 7, we have included a more focused discussion on the advantages and disadvantages of the commonly used methods, explaining why we believe that summarizing the available information is an important first step before making critical evaluations.

Comments 4: Lines 83-95: Add bibliographic support for these ideas 

Response 4: Additional references have been included in the Introduction to better support and justify these ideas. The cited papers provide examples of the usefulness of different types of tests related to the assessment of specific cognitive functions. See Lines 128-152.

Comments 5: Lines 111-112: Adjust font format to what is required 

Response 5: We have adjusted the size of the attached figure and updated the formatting of its caption accordingly. See Lines 167-170.

Comments 6: Line 151: and/or?

Response 6: Yes, the correct form is “and/or” meaning that some abstracts contained all three words, while others included only one. To improve clarity, we revised the sentence keeping the term “and”. See Line 229.

Comments 7: Line 216: There is a lack of discussion about the advantages and disadvantages of the most commonly used methods for measuring cognitive flexibility and memory. 

Response 7: To clarify the discussion on the advantages and disadvantages of the commonly used methods in section 4.2, we have added the following sentence: “The choice largely depends on practical considerations such as available resources, space, and personnel. For example, the use of an arena requires only a relatively simple external structure and minimal staff involvement, whereas problem-solving stations involve multiple objects and require more intensive supervision. Therefore, there is no clear preference for one test over another, and they should ideally be used in combination to achieve a more comprehensive assessment of cognitive abilities (Tanila, 2018; Hunsaker, 2012). Another key consideration is the temperament and personality of the animal: situations involving isolation or restraint should be avoided in animals that are easily stressed or not habituated, as these conditions can introduce clear biases in the results (Finkemeier et al., 2018).”. See Lines 1023-1032. This topic is also discussed throughout several sections of the Results and Discussion. For instance, as noted in the Conclusion, the main advantages of the methods are related to managerial factors, such as including a habituation/training period and using appropriate rewards to motivate the animals.

Comments 8: Line 222: Change “:” to “.”

Response 8: We changed “:” to “.”. See line 901.

Round 2

Reviewer 1 Report

Comments and Suggestions for Authors

There are still major concerns regarding the scope, search strategy and screening strategy as detailed below. We have numbered these consistent with the original report and your responses for ease of tracking:

Major issues

Major point 1:

The relevance of studying cognition in sheep is somewhat clearer. However, the introduction still contains an extensive summary of multiple tests used in mice, without any clear link to the question or rationale for the study. Restriction of the scoping review to studies only conducted in the last 15 years does not follow the intent of a scoping review, since it does not adequately map the body of literature.

Major point 2:

The purpose of a scoping review is not merely for when systematic reviews are not feasible. While scoping reviews can help map and summarize the available literature, they also require some level of appraisal of the identified literature and identification of knowledge gaps.

Major point 3:

Thank you for expanding your sources to include the PubMed database. PubMed has excellent coverage of biomedical literature (which includes many studies of sheep cognition, particularly those where sheep are being utilised as an experimental model to investigate clinical disease). Embase has still not been included and provides more in depth coverage of European literature. The exclusion of pre-2010 studies is not well justified and is contrary to the purpose of a scoping review; merely stating that your scope is recent research does not justify the approach.

The search string is improved but still limited. Broader behavioural terms could have been included in the search as well as terms related to discrimination learning, which is a key executive function.

The fact that you have still restricted your summary to the same 20 studies continues to raise concerns about the selection criteria used. We have several concerns about major flaws in the screening criteria:

  • At the first screening stage, it is not appropriate to exclude articles without an abstract. These must be screened at full-text to assess relevance.
  • Also at abstract stage, requiring these key words will exclude relevant papers that include tests of cognitive function in sheep. It does not make logical sense to perform a “more in depth review of each title” – you cannot obtain the required information from the title alone!
  • Requiring inclusion of a description of the test area (size etc) is not an appropriate exclusion criteria given the stated scope of the review.
  • Similarly, excluding studies with “inclusion of miscellaneous topics unrelated to the objective of the review” will exclude all studies that test cognition AND other outcomes – again not appropriate. As a result, the selection strategy fails to capture studies where cognition is one of several outcomes. A key example of this is the study mentioned in our previous review - Chasles et al. (2024; PMID: 38697024), does indeed include tests of anxiety-like behaviour but also explicitly tests cognitive function in separate tests.

Indeed, the conclusion “This outcome highlights the importance of precise terminology, accurate keyword selection, and clear titles in the original publications, to ensure that relevant research can be effectively identified and included in future evidence synthesis” reflects failures of the screening process, not the original papers!

Major point 4:

It is not clear what “verified by the second author” means in the context of a scoping review. There is also no information provided on conflict resolution for the screening process. In the gold-standard approach to scoping reviews, sources are independently assessed at eah stage of the screening process and any conflicts of opinion then resolved by a third author or collectively by consensus.

Major point 5:

A “practical” framework in cognitive neuroscience should still be conceptually defensible. The current domains remain arbitrary with some miscategorisation still evident in the review (eg. Table 1 associates a two-choice discrimination task with memory).

Major point 6:

Table 1 has been improved, however for consistency, where studies did not report the sex of the animals this should be explicitly stated. If age and sex are important characteristics, then the review should include all studies using a specific apparatus or test, not just the first study in which it is described, since other studies likely include animals of differing ages and sexes.

Major point 7:

While scoping reviews do not require formal critical appraisal they should still contextualise literature quality. There is still minimal discussion around validity of the cognitive tests.

Major point 8:

This point has been addressed satisfactorily.

Major point 9:

The discussion of this issue under section 4.2 Type of test” is improved, however the summary at the start of the review still recommends standardisation of tests.

Minor points for revision

Most minor points have been addressed satisfactorily.

Minor point 7:

Four-lateral arenas is an extremely unclear term!

Major point 8:

Since you have only included papers describing the first use of the test, you haven’t defined what proportion of papers used different types of tests.

Author Response

We greatly appreciate Reviewer 1’s thorough and in-depth analysis, which has substantially improved the clarity and quality of our manuscript.

Major issues

Major point 1: The relevance of studying cognition in sheep is somewhat clearer. However, the introduction still contains an extensive summary of multiple tests used in mice, without any clear link to the question or rationale for the study. Restriction of the scoping review to studies only conducted in the last 15 years does not follow the intent of a scoping review, since it does not adequately map the body of literature.

Response 1: We thank the reviewer for reiterating this important point. To address the concern, we implemented two major clarifications, in accordance with the editor’s suggestion. First, we modified the title of the review to explicitly state that our aim is to examine only the recent literature, thereby making the temporal focus transparent from the outset (See Line 3). Second, regarding the description of murine behavioural tests, our intention is not to provide an exhaustive overview of mouse research but rather to briefly illustrate commonly used paradigms in cognitive testing. Because mice represent the most extensively studied model in behavioural neuroscience, referencing these paradigms helps contextualize the types of cognitive tasks that have informed the development or adaptation of similar tests in sheep. To make this rationale explicit, we added a bridging sentence to clearly present the connection and justify the presence of murine examples in the introduction (See Lines 76-80).

Major point 2: The purpose of a scoping review is not merely for when systematic reviews are not feasible. While scoping reviews can help map and summarize the available literature, they also require some level of appraisal of the identified literature and identification of knowledge gaps.

Response 2: We revised the wording to clarify that a scoping review is not merely an approach used when a systematic review is not feasible. The sentence now reads “and also” instead of “especially” to reflect this (See Line 129). Furthermore, we did not list knowledge gaps in a separate section, as we chose to discuss them within the context of the different setups and cognitive functions investigated. For example, the absence of a single standardized milestone, or the use of varying habituation/training phases prior to testing. As noted in the Limitations section, we acknowledge that a formal qualitative appraisal was not performed, as this was not the primary objective of our review.

Major point 3: Thank you for expanding your sources to include the PubMed database. PubMed has excellent coverage of biomedical literature (which includes many studies of sheep cognition, particularly those where sheep are being utilised as an experimental model to investigate clinical disease). Embase has still not been included and provides more in depth coverage of European literature. The exclusion of pre-2010 studies is not well justified and is contrary to the purpose of a scoping review; merely stating that your scope is recent research does not justify the approach.

The search string is improved but still limited. Broader behavioural terms could have been included in the search as well as terms related to discrimination learning, which is a key executive function.

The fact that you have still restricted your summary to the same 20 studies continues to raise concerns about the selection criteria used. We have several concerns about major flaws in the screening criteria:

  • At the first screening stage, it is not appropriate to exclude articles without an abstract. These must be screened at full-text to assess relevance.
  • Also at abstract stage, requiring these key words will exclude relevant papers that include tests of cognitive function in sheep. It does not make logical sense to perform a “more in depth review of each title” – you cannot obtain the required information from the title alone!
  • Requiring inclusion of a description of the test area (size etc) is not an appropriate exclusion criteria given the stated scope of the review.
  • Similarly, excluding studies with “inclusion of miscellaneous topics unrelated to the objective of the review” will exclude all studies that test cognition AND other outcomes – again not appropriate. As a result, the selection strategy fails to capture studies where cognition is one of several outcomes. A key example of this is the study mentioned in our previous review - Chasles et al. (2024; PMID: 38697024), does indeed include tests of anxiety-like behaviour but also explicitly tests cognitive function in separate tests.

Indeed, the conclusion “This outcome highlights the importance of precise terminology, accurate keyword selection, and clear titles in the original publications, to ensure that relevant research can be effectively identified and included in future evidence synthesis” reflects failures of the screening process, not the original papers!

Response 3: We thank the reviewer for the thorough and detailed comments. As in the first round, we have carefully considered all points raised and have made clarifications and revisions where appropriate. In particular, based on the editor’s guidance, we implemented two major updates:

  • We modified the title of the review to explicitly state that our aim is to examine only the recent literature, making the temporal focus clear from the outset (see Line 3).
  • We revised and expanded the section “Limitations of this scoping review” to incorporate both previously mentioned limitations and those highlighted in Major Point 3. This updated version provides a more comprehensive acknowledgment of the methodological boundaries of our approach (Lines 496–506 & 510-512).

Major point 4: It is not clear what “verified by the second author” means in the context of a scoping review. There is also no information provided on conflict resolution for the screening process. In the gold-standard approach to scoping reviews, sources are independently assessed at eah stage of the screening process and any conflicts of opinion then resolved by a third author or collectively by consensus.

Response 4: We thank the reviewer for requesting clarification. To make the screening and decision-making process more transparent, we have revised the manuscript to provide a detailed description of the roles of the two authors involved. Specifically, the entire screening process was conducted independently by the first author (D.G.) and the second author (E.D.C.) as a double-check. In cases of disagreement, both authors reviewed the relevant studies together and reached a consensus on inclusion or exclusion. See Lines 185-188.

Major point 5: A “practical” framework in cognitive neuroscience should still be conceptually defensible. The current domains remain arbitrary with some miscategorisation still evident in the review (eg. Table 1 associates a two-choice discrimination task with memory).

Response 5: We thank the reviewer for this important clarification regarding the conceptual consistency of the proposed framework. In response, we revised Table 1, Table 2, and the corresponding sections of the Results and Discussion to correct the misclassification identified by the reviewer. Specifically, we agree that the study previously grouped under “memory” is more accurately characterised as assessing only visual (sensory) discrimination. This adjustment improves the coherence of the framework and ensures that each task is aligned with an appropriate cognitive domain.

The revised tables and the updated textual descriptions now reflect this reclassification.

Major point 6: Table 1 has been improved, however for consistency, where studies did not report the sex of the animals this should be explicitly stated. If age and sex are important characteristics, then the review should include all studies using a specific apparatus or test, not just the first study in which it is described, since other studies likely include animals of differing ages and sexes.

Response 6: Firstly, to improve clarity and ensure full consistency, we added a dedicated column reporting the sex of the animals for each study. Although all included papers did specify the sex of the animals (this information is also mentioned in the Results section) it was not consistently presented in the table. The revised version now provides a standardized and easy-to-read format, making the reporting more transparent.

Secondly, we fully agree that sex and age, together with personality and temperament, are important biological characteristics that can influence performance in cognitive tasks. For this reason, we reported these variables in the table to provide readers with contextual information. However, the primary aim of this scoping review was to map and classify the types of executive cognitive tests used in sheep, rather than to compare outcomes across different categories or to evaluate how sex or age modulate task performance. Our intention was not to suggest that a given cognitive function should be assessed only in animals of a specific sex using a particular apparatus. Rather, our aim was to indicate that a certain test has been used to evaluate that cognitive function and, in the corresponding study included in the review, the authors happened to assess it in animals of a given sex. In this framework, the demographic details serve as descriptive context for each test implementation, but they were not used as selection criteria nor as a basis for methodological recommendations.

Major point 7: While scoping reviews do not require formal critical appraisal they should still contextualise literature quality. There is still minimal discussion around validity of the cognitive tests.

Response 7: We sincerely thank the reviewer for this valuable observation and for encouraging us to reflect further on the context of literature quality. We fully acknowledge that, although formal critical appraisal is not a requirement for scoping reviews, providing some discussion on the validity of the included cognitive tests can help readers interpret the findings more accurately.

To address this, we have added brief clarifications in the Results section noting that most included studies were published in peer-reviewed journals and retrieved through multiple databases (Scopus, PubMed, and CAB Abstracts), and that many of the tasks have been previously employed and validated in previous work. This indicates that the tests are generally robust and appropriate for assessing executive cognitive functions in sheep. See Lines 238-239 and 246-249.

Major point 8: This point has been addressed satisfactorily.

Response 8: We thank the reviewer for acknowledging that this point, raised during the first round of revisions, has now been satisfactorily addressed.

Major point 9: The discussion of this issue under section 4.2 Type of test” is improved, however the summary at the start of the review still recommends standardisation of tests.

Response 9: We thank the reviewer for highlighting this point. To address the concern, we revised the introductory summary to ensure that it no longer implies a recommendation toward standardising cognitive tests. Specifically, we removed the previous sentence suggesting that the use of uniform testing methods would enhance reliability and comparability. In its place, and to maintain consistency with Section 4.2, we incorporated a more neutral and descriptive statement indicating that the review showed that certain executive cognitive functions tend to be associated with specific task designs. This revision aligns the summary with the intended scope of the review and avoids unintended prescriptive interpretations. See lines 17-18.

Minor points for revision

Minor point 7: Four-lateral arenas is an extremely unclear term!

Response 10: We changed “four-lateral” in “quadrilateral”. See Lines 327, 451 and 461.

Major point 8: Since you have only included papers describing the first use of the test, you haven’t defined what proportion of papers used different types of tests.

Response 11: We thank the reviewer for the comment and hope we have correctly understood it. Few papers included more than one test, usually combining executive cognitive functions (ECF) and emotionality tasks (e.g., McBride et al., 2018, doi:10.1007/s00221-018-5370-8). In cases where multiple tests were reported, we focused on the ECF tasks and included them only at their first mention. For example, McBride et al., 2018, used the same arena described previously in 2016; therefore, this study was not included among the 20 papers for that task, but this decision was not influenced by the presence of other tests. Generally, the papers we included used a single test, and in cases of multiple tests, we included them only if they introduced a novel paradigm (e.g., McBride et al., 2015, with mirror-based setups, doi:10.1007/s10071-014-0807-3).

Reviewer 2 Report

Comments and Suggestions for Authors I thank the authors for taking the suggestions into account and making the text clearer.

Author Response

We sincerely thank the reviewer for their positive feedback and for acknowledging the clarifications made. We greatly appreciate the effort and time spent in reviewing our manuscript.